# Family Structure and Maternal Depressive Symptoms: A Cross-National Comparison of Australia, the United Kingdom, and the United States

Kirsten Rasmussen [1,*], Elizabeth K. Sigler [1], Sadie A. Slighting [1], Jonathan A. Jarvis [1], Mikaela J. Dufur [1] and Shana Pribesh [2]

1 Department of Sociology, Brigham Young University, 2008 JFSB, Provo, UT 84602, USA; elizabethmarie.sigler@gmail.com (E.K.S.); sadie.slighting@gmail.com (S.A.S.); jonathan_jarvis@byu.edu (J.A.J.); mikaela_dufur@byu.edu (M.J.D.)
2 Department of Educational Foundations and Leadership, Old Dominion University, 2307 Education Building, Norfolk, VA 23529, USA; spribesh@odu.edu
* Correspondence: kirsten.rasmussen2153@gmail.com

**Abstract:** The purpose of this study is to understand the relationship between family structure and maternal depressive symptoms (MDS) in Australia, the United Kingdom, and the United States. Family structures that involve transitions across life's course, such as divorce, can alter access to resources and introduce new stressors into family systems. Using the stress process model, we examine the links between family structure, stress, resources, and MDS. Using nationally representative data from Australia, the United Kingdom, and the United States and cross-sectional models for each country, we find that family structure may influence MDS differently in the UK than it does in Australia or, especially, the US. Specifically, mothers in the UK who either enter or leave a marriage after the birth of their child experience increased levels of MDS compared with mothers who do not experience a similar transition. These findings demonstrate that the effects of family structure transitions across life's course may vary according to the country context as well as to the mother's access to resources and exposure to stress. Considering that the effects of family structure transitions are not universal, this indicates that greater attention should be paid to the country contexts families exist in and the effects that public policies and social safety nets can have on MDS.

**Keywords:** depression; family structure; family transitions; mental health; stress





## 1. Introduction

According to the World Health Organization (2017), depression affects about 6% of American adults each year, which is similar to the rates of depression in other developed, high-income nations. Among parents in western high-income nations, depression is more common among mothers than fathers, as women are almost twice as likely as men to develop a major depressive disorder (Kessler et al. 1994; Mackinnon et al. 2004; Richards 2011). Depression might have particularly far-reaching effects in the family system because of its consequences not only for the mental health trajectory of mothers but also for their partners and children (Avison 2016; Rehman et al. 2015). For example, depressed mothers report higher levels of marital conflict and are generally less engaged with their children compared with nondepressed mothers (Burke 2003; Goodman et al. 2011). Although the literature on how maternal depression relates to child trajectories is robust, relatively few studies have examined the family mechanisms that contribute to maternal depression in the first place (Cooper et al. 2009; Osborne et al. 2012). Family structure transitions such as marriage or divorce can be sources of stress and instability, and there is evidence that these changes can affect multiple parts of the family system, including children, in many ways (Amato and

Hohmann-Marriott 2007). However, further research is needed to better understand how these changes affect maternal mental health (Brown 2004; Brown et al. 2015).

Furthermore, though we have evidence that depression rates are similar across high-income western nations, little research has looked into how any relationship between family structure and maternal depression may or may not be specific to a country's cultural context (Bromet et al. 2011). Australia, the United Kingdom, and the United States are all high-income anglophone nations that share similar western capitalist values, yet there are important cultural and political differences among these three countries that may influence any associations between family structure and maternal depression (Parcel et al. 2012). For example, the divorce rate is much higher in the US than it is in Australia or the UK (Australian Bureau of Statistics 2021; CDC/NCHS 2019; Eurostat 2021), indicating that there are cultural differences in the prevalence and perception of different family structures that might affect the mental health outcomes of mothers in certain family structures (Pryor 2013). Furthermore, research has found that parents in the US experience significantly lower levels of happiness than nonparents do, whereas Australia and the UK do not report similar happiness gaps, perhaps because their public policies better support families where parents are married or living with their partner (Glass et al. 2016). Of these countries, the US is the only nation that does not offer universal maternity leave, and Australia and the US spend less per capita on childcare services than the UK does (Daly and Ferragina 2018; Thévenon 2011). It therefore becomes important to acknowledge the broader country context when exploring the relationship between family structure and maternal depression because of the role that a country's culture or social safety nets may play in mitigating or exacerbating the influence of stress associated with family transitions on a mother's well-being. Using the stress process model, we examine how the association between family structures or transitions and maternal depressive symptoms may vary across three culturally similar countries: Australia, the United Kingdom, and the United States. We model the cross-sectional relationships between family structures, stressors, resources, and the demographic variables for each country and conclude that the relationship between family structure and maternal mental health does vary across country, with family structure influencing maternal depressive symptoms quite differently in the UK than they do in Australia or the US.

## 2. Literature Review

### 2.1. Stress Process Model

To contextualize the relationship between family structure and maternal depressive symptoms, we turned to the stress process model. This conceptual framework includes three major components: stressors, moderators or mediators, and mental health outcomes (Pearlin et al. 1981; Pearlin and Bierman 2013). Stressors may be either chronic strains or general life events that occur across course of life and may be specific to an individual's roles, such as parenting role strains. Parental role strains refer to the challenges and problems that parents experience which may have ramifications for not only the parent's mental health but also their child's well-being (Nomaguchi and Milkie 2017). Resources that help individuals cope with stress can intervene in this process by moderating or mediating the effects of the stressor on mental health outcomes. Moderating resources can refer to a variety of supports an individual has access to, including coping skills, social supports, or financial resources (Pearlin and Skaff 1996). Additionally, broader policies and institutions that have implications for parents, such as workplace or state policies that support families and reduce the burden of parenting, can similarly help alleviate the effects of stressors on individuals (Nomaguchi and Milkie 2017, 2020; Pearlin and Bierman 2013). Because resources may be unequally distributed across populations and may vary across an individual's life course, accounting for access to financial resources, policy supports, and the social contexts in which they occur is crucial to understanding the manifestation of depressive symptoms in mothers (Pearlin et al. 2005).

We introduce two contexts in which to test the application of the stress process model: country context and family structure. Because every aspect of the stress process model is influenced by an individual's various social statuses, the country and family contexts become important for understanding the processes that link stress with maternal depressive symptoms (McLeod 2012; Nomaguchi and Milkie 2020). To explore the potential importance of the country context, we took a cross-national approach by comparing mothers in Australia, the United Kingdom, and the United States. These countries are all English-speaking, high-income, western nations that share a common political heritage (Parcel et al. 2012). However, there are notable cultural and political differences across these countries, including variations in approaches to social support. For example, the US consistently spends less on housing and family benefits compared with Australia or the UK, and both Australia and the US invest less in childcare services than the UK does (Daly and Ferragina 2018; Thévenon 2011; Valente 2019). These cultural differences indicate that there are important macro-level distinctions in support of families across these countries that could affect whether mothers have enough coping resources to handle the strain of being a parent or the strain of experiencing family transitions, indicating that the link between family structure and maternal depressive symptoms may differ according to the country context in which the mother resides.

The second context we examined was the family structure itself, which we measured using eight categories that considered the mother's union status, the nature of their partner's relationship to the child, and whether this relationship was stable over time. A mother's ability to cope with stress may vary according to their family structures, as the family structures they reside in will influence how much role strain they experience and what level of financial support they have. For example, mothers who reside in a stable union likely have relatively lower levels of parental strain, as they are able to share the parenting burden with a partner, as well as also having greater access to financial resources they can use to cope with the demands of being a parent. Conversely, single mothers may experience heightened levels of role strain as they parent on their own, combined with limited access to resources because they are unable to rely on a partner for financial or social support. We might then expect mothers who do not reside in married stable families to experience increased levels of depressive symptoms, as mothers in these alternative family structures may experience greater exposure to and difficulty coping with stress. Using the stress process model will allow us to better understand the processes that link family structure and maternal depressive symptoms, while also testing the applicability of the stress process model in the new contexts of family structure and country context.

*2.2. Stress and Maternal Depressive Symptoms*

Mothers who have experienced a family structure transition are likely to experience heightened stress and increased depressive symptoms, owing to the transitions in their parental role trajectories and the uncertainty associated with these changes (Macmillan and Copher 2005; Pace and Shafer 2015). This is especially true for nonbiological parents, who have been found to experience more depressive symptoms and parenting stress than biological parents (Shapiro and Stewart 2011). This indicates that the family structure a mother resides in is important for understanding the manifestation of depressive symptoms. Conversely, residing in a family structure that follows a transition, such as a post-birth union or divorce, may be associated with decreased stress levels. Mothers who begin to cohabit with the biological father of their child experience less parental strain than single mothers who continue to live alone, and mothers in high-distress marriages experience higher levels of happiness after divorce (Amato and Hohmann-Marriott 2007; Cooper et al. 2009). Union formation may also protect against mental or physical health problems that are associated with increased stress, as well as introducing new resources and supports that mothers have access to (Cooper et al. 2009; Osborne et al. 2012). While there is evidence that union status is related to an individual's depression levels, it is less clear how the role of being a parent affects this association (Hollist et al. 2007). Accounting for both

an individual's marital and parental status allowed us to capture a more comprehensive view of the family structure, especially in regard to the sequencing of transitions in these roles, which can have consequences on the mental health of mothers (Avison et al. 2008; Macmillan and Copher 2005). Embedding family structure in a life course context allowed us to better distinguish the pathways in which stress was most strongly associated with the manifestation of maternal depressive symptoms. By considering the union status, parental role, and order of these life events, we hope to explore how these social roles together shape an individual's exposure to stress and the level of depressive symptoms.

### 2.3. Resources and Maternal Depressive Symptoms

Resources, or a lack of them, might also help explain the poorer maternal mental health outcomes that are found across family structures, particularly in how resources are linked to the stress process. Some family structures and transitions may dilute family resources, including physical, financial, and social resources (Osborne et al. 2012). For example, mothers facing a divorce or separation may experience a decline in the resources and support they have access to and therefore an associated increase in stress (Osborne et al. 2012). As individuals move through the course of their lives, the social relationships they are able to draw support from change, and transitions in family structure may precipitate the loss of these social supports (Pearlin and Skaff 1996). For example, residential moves often accompany a divorce or separation and may disrupt mothers' social networks, decreasing the resources and support they are able to rely on (Beck et al. 2010). Furthermore, evidence suggests that divorced families have less social capital than families with continuously married parents, and single mothers are likely to perceive that they have less social support than married mothers (Cairney et al. 2003; Sun and Li 2007).

Single mothers often report experiencing heightened stress, owing to their housing instability and decreased financial support (Brown 2004). Furthermore, single parents have been found to report the lowest levels of emotional well-being among parents, indicating that the negative effects of stress associated with parenthood may be more acute for certain family structures (Collins and Glass 2018). Given that financial instability is generally associated with adult depression (Zimmerman and Katon 2005), we would expect that financial strains that result from certain family structures or family structure instability would also be related to maternal depressive symptoms. At the same time, however, family structure changes that occur through entering rather than leaving a partnership may consolidate resources across multiple actors, leading to an increase in the resources and support a mother has access to. If this is the case, entrance into a marriage might be associated with a decrease in depressive symptoms (Lamb et al. 2003).

The stress associated with limited financial resources may be particularly important in understanding the manifestation of maternal depressive symptoms. The stress process model, coupled with the life course perspective, indicates that the strain of continued economic deprivation may have a particularly acute effect on individuals' mental health (Pearlin et al. 2005). This occurs through not only the economic strain itself but also through the concomitant stressors that accompany the stress of financial hardship (Pearlin et al. 2005). Mothers who reside in family structures associated with economic disadvantage, such as single mothers, therefore often experience elevated levels of stress and role strain, indicating that the family structure context a mother resides in is an important factor in understanding the manifestation of maternal depressive symptoms (Avison 2009).

Collins and Glass (2018) found that public policies that alleviate the economic strain associated with childcare may be the most effective way to improve the well-being of mothers, indicating that the broader cultural context of a country is related to a family's access to resources and is an important aspect to understanding maternal depression symptoms. For example, Australia and the United Kingdom spend more per capita on families through public policies than the United States does, suggesting that a mother's financial security and associated stress will vary by country context (Glass et al. 2016). Overall, resources are an important component of the stress process, and understanding the

distribution of resources across both the family structure context and the national context is therefore an essential aspect to understanding whether these contexts may be associated with maternal depressive symptoms.

If we find that stress is among the most important predictors of depressive symptoms across each of the countries examined here, we will have evidence that stressors are universal (or at least common factors) and not dependent on context-specific social and financial safety nets and that we can understand the stress process in similar ways in different country contexts. However, if we find this pattern is not uniform across countries, particularly as it relates to the way resource allocations lead to maternal stress, we will have evidence instead that the association is dependent on the country context and is perhaps specific to the ways political systems in each country distribute resources. The same is true for family structure. If we find that stress is increased in family structures that are associated with lost resources but ameliorated in family structures that bring additional physical and social resources into the family system, this will suggest a broad application of the stress process. Our research will facilitate a deeper understanding of the process through which stress is associated with maternal depressive symptoms and also allow us to test the utility of the stress process model in different family structures and country contexts.

## 3. Methods

### 3.1. Nations in the Analysis

The three countries in our analysis are all English-speaking countries with similar colonial histories and roots and are similar across a number of important social and economic measures (see Table 1). Each of these nations are advanced economies, with a Gross Domestic Product (GDP) among the top 15 nations in the world (World Bank 2021b). The pre-COVID pandemic 2019 unemployment rates were relatively similar, ranging from 3.7% in the US and UK to 5.2% in Australia (World Bank 2021c). Inequality in these three nations varies from more egalitarian in Australia (GINI score of 32.5 and closer to the OECD average of 31.3) (OECD 2021a) to more unequal nations like the United Kingdom and the United States (GINI scores of 36.6 and 39.5, respectively) (OECD 2022a). These are also highly educated nations. Among the 25–34-year-olds in each country in our analysis, a higher percentage have college degrees than the OECD average of 45.5% (OECD 2022b). Marriage in Australia and the UK occurs at a similar age to the OECD average of 30.7 for women and 32.8 for men (OECD 2021b), while first marriages occur 2–3 years earlier for those in the US (30.4 for men and 28.6 for women) (U.S. Census Bureau 2021). These nations also have similarly low fertility rates (all below 2 per 1000) (World Bank 2020a) and crude divorce rates ranging from below 2 per 1000 in the UK and Australia (1.8 and 1.9, respectively) to as high as 2.9 in the US (Australian Bureau of Statistics 2021; CDC/NCHS 2019; Eurostat 2021). These statistics suggest similar movement through the second demographic transition in each of the three nations in our analysis, while the earlier marriage ages and higher divorce rates in the US indicate important nation-specific variations for this transition.

**Table 1.** National indicators.

| | GDP Rank (2020) | GDP per Capita (2020) | Unemployment Rate (2019) | GINI (Year) | Population with Tertiary Degree (Age 25–34) | Age at 1st Marriage (M/F) | Fertility Rate (2019) | Crude Divorce Rate (per 1000) |
|---|---|---|---|---|---|---|---|---|
| Australia | 13 | 51,692.8 | 5.2% | 32.5 (2018) | 54.6% | 32.2/30.6 | 1.66 | 1.9 (2020) |
| UK | 5 | 41,124.5 | 3.74% | 36.6 (2019) | 55.8% | 33.4/31.5 | 1.65 | 1.8 (2016) |
| USA | 1 | 63,413.5 | 3.67% | 39.5 (2019) | 51.9% | 30.4/28.6 | 1.71 | 2.9 (2018) |

Sources: (Australian Bureau of Statistics 2021; CDC/NCHS 2019; Eurostat 2021; OECD 2021a, 2021b, 2022a, 2022b; Office of National Statistics 2019; U.S. Census Bureau 2021; World Bank 2020a, 2021b, 2021c, 2021d). Notes: GDP per capita in USD. GINI index from the most recent available year. US and Australia report median age at first marriage, UK reports mean age at first marriage.

### 3.2. Data

Our analyses used longitudinal data sets from Australia, the United Kingdom, and the United States. Our Australian data came from Growing Up in Australia: The Longitudinal Study of Australian Children, which commenced in 2003, starting at targeting children's infancy and following the participants and their parents approximately every 2 years since (Australian Institute of Family Studies 2003). The data for the United Kingdom came from the UK Millennium Cohort Study, which tracks children born between 2000 and 2002 approximately every 3 years (Centre for Longitudinal Studies 2000). Our United States data came from the US Early Childhood Longitudinal Study, Kindergarten Class of 1998–99, which follows children from their entry into kindergarten up through the eighth grade (National Center for Education Statistics 1998). We selected the fourth wave or sweep from each data set, which corresponded to a child age that averaged around 7 years old, resulting in a population of 4242 mothers in Australia, 13,857 mothers in the UK, and 9604 mothers in the United States. We used retrospective data from parent respondents about family structure, allowing us to distinguish between stable and transitory family structures. Our data came from questionnaires administered to the main parent (overwhelmingly mothers) in each country. Because of our focus on maternal mental health, we omitted a small proportion of cases where only fathers provided information (AUS: 2.59%; UK: 0.06%; US: 6.01%). For all data sets, we limited our sample to mothers for whom we could identify measures of family structure across waves. This resulted in a total sample size of 3972 mothers from Australia, 12,821 mothers from the UK, and 8624 mothers from the US. We note that there were too few respondents who reported being in a same-sex romantic or parenting relationship in the data to be able to make efficient estimates for this group. In addition, the data did not allow us to distinguish if mothers in single-parent families identified as members of the LGBTQIA+ community. As a result, all coupled mothers in our data were in opposite-sex relationships. We acknowledge that patterns may differ for parents in same-sex relationships or mothers who are members of a sexual or gender minority group.

### 3.3. Measures

Concerning maternal depressive symptoms (MDS), in each data set, we constructed a scale capturing psychological distress (hereafter referred to as an MDS scale). In the US data, our scale came from adding together the 12 questions found in the Center for Epidemiological Studies Depression Scale (CES-D). For our MDS scales in Australia and the UK, we added together the six questions found in the six-item version of the Kessler Psychological Distress Scale. The questions found in these scales were coded on a 5-point scale, ranging from *none of the time* (0) to *all of the time* (4) and indicating how often the mother experienced symptoms of psychological distress, such as "How often do you feel hopeless?" The scales in our data ranged from 3–30 in Australia ($\alpha = 0.86$), 1–25 in the UK ($\alpha = 0.88$), and 0–36 in the US ($\alpha = 0.91$). The MDS scale in each country was then standardized, with positive values indicating above-average levels of MDS (see Table 2 for descriptions for all variables used in the analyses, reported for each country).

Family Structure: We constructed the family structure through parent data that included the mother's relationship to the child, the mother's marital status, the relationship of the mother's partner (if applicable) to the child, and information from household rosters (see Augustine and Kimbro 2013 for similar approaches). Due to data limitations, we focused on heterosexual partners in our analysis. Transitions in marital and parental status through the course of life formed distinct trajectories for mothers, which is why we employed eight unique family structures that captured stability and the timing of transitions (Avison et al. 2008). Three of our family structure categories captured stability in family structure over time:

**Table 2.** Descriptive statistics for Australia, the United Kingdom, and the United States.

| Variable | Proportion or *M* | | | *SD* | | | Range |
| | Australia | UK | US | Australia | UK | US | Australia, UK, US |
|---|---|---|---|---|---|---|---|
| Family Structure | | | | | | | 1–8 |
|   Biological Married Stable | 0.643 | 0.482 | 0.603 | | | | |
|   Biological Cohabiting Stable | 0.089 | 0.069 | 0.016 | | | | |
|   Biological Single Stable | 0.056 | 0.081 | 0.072 | | | | |
|   Post-Birth Biological Married | 0.056 | 0.114 | 0.095 | | | | |
|   Post-Birth Stepfamily | 0.015 | 0.029 | 0.055 | | | | |
|   Post-Birth Biological Cohabiting | 0.015 | 0.036 | 0.019 | | | | |
|   Post-Birth Social Family | 0.035 | 0.022 | 0.026 | | | | |
|   Post-Birth Transition to Single | 0.091 | 0.167 | 0.113 | | | | |
| Maternal Depressive Symptoms | 0.062 | −0.006 | 0.028 | 0.021 | 0.011 | 0.017 | |
| Family Stressor Scale | 0.003 | 0.044 | 0.058 | 0.019 | 0.012 | 0.025 | |
| Income (reported in quintiles) | | | | | | | 1–5 |
|   Bottom | 0.242 | 0.193 | 0.338 | | | | |
|   Second | 0.202 | 0.194 | 0.129 | | | | |
|   Third | 0.196 | 0.201 | 0.297 | | | | |
|   Fourth | 0.182 | 0.205 | 0.113 | | | | |
|   Top | 0.178 | 0.206 | 0.123 | | | | |
| Moved | 0.263 | 0.111 | 0.195 | 0.008 | 0.003 | 0.008 | 0–1 |
| State Support | 0.550 | 0.122 | 0.037 | 0.009 | 0.004 | 0.003 | 0–1 |
| Mother's Employment | | | | | | | 1–3 |
|   Full-time | 0.245 | 0.142 | 0.463 | | | | |
|   Part-time | 0.393 | 0.482 | 0.233 | | | | |
|   Not in paid labor force | 0.362 | 0.377 | 0.304 | | | | |
| Home Ownership | 0.696 | 0.646 | 0.714 | 0.009 | 0.005 | 0.008 | 0–1 |
| Household Size | 4.574 | 3.454 | 4.619 | 0.024 | 0.013 | 0.022 | 2–13, 2–19, 2–15 |
| Mother's Education | | | | | | | 1–5 |
|   Less than secondary school | 0.111 | 0.164 | 0.116 | | | | |
|   Secondary school | 0.305 | 0.543 | 0.256 | | | | |
|   Some college | 0.159 | 0.113 | 0.355 | | | | |
|   Post-secondary degree | 0.229 | 0.139 | 0.179 | | | | |
|   Higher degree | 0.201 | 0.042 | 0.095 | | | | |
| Parent's Immigration Status | | | | | | | 0–1 |
|   Neither parent is an immigrant | 0.591 | 0.685 | 0.806 | 0.009 | 0.004 | 0.007 | |
|   One or more parent is an immigrant | 0.409 | 0.315 | 0.194 | 0.009 | 0.004 | 0.007 | |
| Child Gender (1 = Male) | 0.509 | 0.512 | 0.508 | 0.009 | 0.005 | 0.008 | 0–1 |
| Child Age (months) | 81.930 | 86.806 | 86.789 | 0.063 | 0.031 | 0.068 | 73–93, 76–98, 76–102 |
| Child Race | | | | | | | N/A, 1–4, 1–5 |
|   White | | 0.865 | 0.603 | | | | |
|   Black | | 0.031 | 0.122 | | | | |
|   Hispanic | | | 0.208 | | | | |
|   Asian | | 0.067 | 0.023 | | | | |
|   Other | | 0.037 | 0.044 | | | | |
| Mother's Age at Birth | 30.632 | 28.523 | 23.988 | 0.103 | 0.063 | 0.086 | 15–48, 14–51, 12–46 |

Note: Australia N = 3972; United Kingdom N = 12,821; United States N = 8624.

     Biological Married Stable: The mother was married to the child's biological father prior to the child's birth and remained married through all subsequent waves.

     Biological Cohabiting Stable: The mother was unmarried but cohabiting with the biological father prior to the child's birth and remained so in all subsequent waves.

Biological Single Stable: The mother was single at the child's birth and neither married nor cohabited since the child's birth.

The remaining five categories captured change or instability in the family structure:

Post-Birth Biological Married: The mother married the biological father after the child was born and remained married through the wave from which we drew the dependent variable.

Post-Birth Stepfamily: The mother married a nonbiological parent after the child was born and remained married through the wave from which we drew the dependent variable.

Post-Birth Biological Cohabiting: The mother began cohabiting with the biological father after the child was born and remained cohabiting with this partner through the wave from which we drew the dependent variable.

Post-Birth Social Family: The mother began cohabiting with a nonbiological parent after the child was born and remained cohabiting with this partner through the wave from which we drew the dependent variable.

Post-Birth Transition to Single: The mother became single after the child's birth. This included mothers who were divorced, separated, or widowed.

For stress, we also included a scale tapping the stressors the mother may have experienced, though there were important differences in how this scale was measured in each country. In Australia, this came in the form of a stress index that measured 23 stressors a mother may be exposed to, such as experiencing a major financial crisis or change in job status. This scale ranged from 0 to 23 ($\alpha = 0.61$). In the UK and US data, there was no equivalent scale, so in each country we created an index from the available data that included two binary variables capturing potential stressors. These variables were chosen because they most closely resembled components of the stressor scale used in the Australian data. In the UK, these stressful scenarios included being behind on household bills (0 = *not behind*, 1 = *behind*) or a partner having a chronic illness (0 = *no*, 1 = *yes*). In the US, the stressors included being evicted (0 = *not evicted*, 1 = *evicted*) or experiencing a death in the family (0 = *no*, 1 = *yes*). Each scale or index was then standardized to allow for better comparisons across countries.

As for other explanatory variables, in addition to our measures of family structure and stress, we also included a theoretical block of variables that captured a family's financial resources which might have explained associations between the family structure and MDS. These variables included income, home ownership, residential mobility, mother's employment, state support, and household size. A less-privileged status for each of these variables was associated with heightened stress and increased MDS levels (Reading and Reynolds 2001; Sheppard 1997). Income was measured in quintiles derived from the entire sample distribution for each individual country (so the lowest Australian quintile was compared with the highest Australian quintile, etc.) (1 = *lowest*, 5 = *highest*). Home ownership and residential mobility were both dichotomous variables that measured whether the family owned their home (0 = *did not own*, 1 = *owned*) and whether they had moved since the previous wave of the study (0 = *had not moved*, 1 = *moved*). The mother's employment was a three-category variable that indicated the employment status of the mother (1 = *full-time*, 2 = *part-time*, 3 = *not in paid labor force*). State support was a binary variable that measured whether the family received income support from the government (0 = *did not receive support*, 1 = *received support*). Household size was a count of the number of people living in the mother's home at the time of the survey.

We also controlled for a block of demographic variables that might help explain the relationship between the family structure and MDS. These variables included the mother's education, parent's immigration status, child gender, child age, and mother's age at birth, all of which have been shown to be related to the prevalence of MDS (Miszkurka et al. 2010; Reading and Reynolds 2001). The mother's education was a categorical variable measuring the highest level of education the mother achieved (1 = *less than secondary school*, 2 = *secondary school*, 3 = *some college*, 4 = *first post-secondary degree*, 5 = *higher degree*). Immigration status was a binary variable that measured whether the mother or their partner (if applicable) were immigrants (0 = *neither was an immigrant*, 1 = *one or more*

*was an immigrant*). Child race was a categorical variable (AUS: no measures of race; UK: 1 = *White*, 2 = *Black*, 3 = *Asian*, and 4 = *Other*; US: 1 = *White*, 2 = *Black*, 3 = *Hispanic*, 4 = *Asian*, and 5 = *Other*), child gender was a binary variable (0 = *female*, 1 = *male)*, and child age measured in months how old the child was at the time of the survey. The mother's age at birth was the age of the mother in years when the child was born.

*3.4. Analytic Plan*

We first reported the descriptive statistics for the sample population in each country. We then ran a cross-sectional nested OLS regression with theoretical blocks to determine the significance of the effect of the family structure, including structures that had been created by previous transitions, on MDS. The first model consisted of only the family structure and MDS, the second added in demographic controls, the third added stressors, the fourth removed stressors and added in our resource variables, and the fifth and final model included the family structure, demographic control variables, stressors, and resources. One approach to our question would be to run multilevel models where country was the second-level indicator; however, because our data were gathered through separate surveys with differences in designs, we were concerned that such an approach could conflate the design effects with what would appear to be the country effects. As a result, we ran the models separately for each country. While we argue that comparing across general patterns across countries is appropriate, we do encourage appropriate caution and understanding of specific coefficients and comparisons. We used appropriate survey weights provided by the data sets in each country. To account for missing data, we used multiple imputation by constructing 20 imputed data sets using the chained equations method in Stata 15 to improve the power and efficiency (Graham 2009). The highest rate of missing data in each country was 21% in Australia, 17% in the UK, and 23% in the US. Post-imputation diagnostics suggested appropriate imputation values.

## 4. Results

We found that most mothers in each country resided in a stable married family, with Australia and the US reporting a higher proportion of mothers in this category than the UK. The second-most-common family structure was post-birth transition to single, with the UK reporting more mothers in this family structure than Australia and the US. The proportions across other family structures were small, with minor differences across countries. For example, the US had fewer stable biological cohabiters than Australia and the UK but more post-birth stepfamilies.

*4.1. Model 1: Family Structure in the Bivariate*

Our first regression model regressed only the Maternal Depressive Symptoms scale on the family structure. We found that the mothers in all alternative family structures in the UK reported higher average levels of MDS compared with mothers in married stable families (Table 3). This pattern was slightly weaker in Australia (Table 4) and the US (Table 5), with only around half of the family structures presenting a similar pattern. In Australia, only the mothers in cohabiting stable, single stable, post-birth social, and post-birth transition to single families were predicted to have significantly higher rates of MDS than married stable mothers. Similarly, in the US, only mothers in single stable, post-birth biological cohabiting, and post-birth transition to single families were predicted to have significantly higher rates of MDS compared with their married stable counterparts.

Because of the prominent role that selectivity factors, resources, and stress occupy in the stress process model, we ran additional models introducing theoretical blocks concerning additional factors to more closely explore the stress process model. We split these predictors into three blocks to tease out their effects on MDS. Our first block model added in the selectivity factors, our second block model included the selectivity factors and the family stress scale, our third block model included the selectivity factors and variables capturing a family's resources, and the final model included all variables.

**Table 3.** Ordinary least squares regression of maternal depressive symptoms regressed on family structure, stress, resources, and controls in the United Kingdom (N = 12,821).

| | Model 1 | Model 2 | Model 3 | Model 4 | Model 5 |
|---|---|---|---|---|---|
| **Family Structure** | | | | | |
| Biological Cohabiting Stable | 0.122 ** | 0.113 ** | 0.095 * | 0.071 | 0.070 |
| | (0.035) | (0.036) | (0.030) | (0.026) | (0.023) |
| Biological Single Stable | 0.449 *** | 0.366 ** | 0.289 | 0.097 | 0.101 |
| | (0.118) | (0.099) | (0.078) | (0.020) | (0.020) |
| Post-Birth Biological Married | 0.140 *** | 0.126 *** | 0.108 ** | 0.106 ** | 0.097 ** |
| | (0.043) | (0.040) | (0.034) | (0.034) | (0.030) |
| Post-Birth Stepfamily | 0.346 *** | 0.311 *** | 0.229 ** | 0.196 * | 0.154 * |
| | (0.064) | (0.060) | (0.043) | (0.042) | (0.031) |
| Post-Birth Biological Cohabiting | 0.305 *** | 0.247 *** | 0.193 ** | 0.134 * | 0.121 |
| | (0.062) | (0.0552) | (0.044) | (0.035) | (0.030) |
| Post-Birth Social Family | 0.213 * | 0.177 | 0.136 | 0.063 | 0.06 |
| | (0.032) | (0.030) | (0.024) | (0.015) | (0.014) |
| Post-Birth Transition to Single | 0.437 *** | 0.398 *** | 0.322 *** | 0.187 *** | 0.176 *** |
| | (0.151) | (0.142) | (0.111) | (0.059) | (0.052) |
| **Mother's Education** | | | | | |
| Secondary school | | −0.201 *** | −0.175 *** | −0.086 * | −0.090 * |
| | | (−0.103) | (−0.088) | (−0.046) | (−0.046) |
| Some college | | −0.289 *** | −0.236 *** | −0.103 * | −0.102 * |
| | | (−0.094) | (−0.076) | (−0.034) | (−0.032) |
| Post-secondary degree | | −0.329 *** | −0.261 *** | −0.108 * | −0.100 * |
| | | (−0.120) | (−0.094) | (−0.040) | (−0.035) |
| Higher degree | | −0.240 *** | −0.177 *** | 0.010 | 0.008 |
| | | (−0.052) | (−0.039) | (0.001) | (0.001) |
| **Parent's Immigration Status** | | | | | |
| One or more parent is an immigrant | | 0.051 | 0.052 | 0.038 | 0.042 |
| | | (0.021) | (0.021) | (0.014) | (0.017) |
| **Child Gender (1 = Male)** | | 0.031 | 0.024 * | 0.030 | 0.024 |
| | | (0.022) | (0.019) | (0.022) | (0.020) |
| **Child Age** | | 0.003 | 0.002 | 0.003 | 0.002 |
| | | (−0.007) | (0.003) | (0.009) | (0.005) |
| **Child Race** | | | | | |
| Black | | −0.015 | −0.071 | −0.086 | −0.116 |
| | | (0.004) | (−0.009) | (−0.008) | (−0.016) |
| Asian | | 0.234 *** | 0.233 *** | 0.161 ** | 0.175 ** |
| | | (0.067) | (0.063) | (0.047) | (0.048) |
| Other | | 0.151 * | 0.101 * | 0.109 | 0.077 |
| | | (0.030) | (0.022) | (0.022) | (0.017) |
| **Mother's Age at Birth** | | −0.001 | 0.000 | 0.005 * | 0.004 * |
| | | (−0.001) | (0.004) | (0.036) | (0.030) |
| **Family Stressor Scale** | | | 0.193 *** | | 0.168 *** |
| | | | (0.222) | | (0.196) |
| **Income** | | | | | |
| Second | | | | −0.031 | −0.017 |
| | | | | (−0.011) | (−0.004) |
| Third | | | | −0.067 | −0.040 |
| | | | | (−0.035) | (−0.022) |
| Fourth | | | | −0.140 ** | −0.094 * |
| | | | | (−0.064) | (−0.046) |
| Top | | | | −0.262 *** | −0.203 *** |
| | | | | (−0.011) | (−0.084) |
| **Moved** | | | | 0.049 | 0.045 |
| | | | | (0.013) | (0.011) |
| **State Support** | | | | 0.296 *** | 0.259 *** |
| | | | | (0.102) | (0.089) |
| **Mother's Employment** | | | | | |
| Part-time | | | | −0.021 | −0.013 |
| | | | | (−0.011) | (−0.009) |
| Not in paid labor force | | | | 0.100 ** | 0.082 * |
| | | | | (0.059) | (0.049) |

**Table 3.** *Cont.*

|  | **Model 1** | **Model 2** | **Model 3** | **Model 4** | **Model 5** |
|---|---|---|---|---|---|
| Home Ownership |  |  |  | −0.110 *** | −0.056 |
|  |  |  |  | (−0.485) | (−0.023) |
| Household Size |  |  |  | 0.006 | 0.009 |
|  |  |  |  | (−0.007) | (−0.005) |
| Adjusted R-squared | 0.032 | 0.053 | 0.098 | 0.083 | 0.116 |
| F-statistic | 57.58 *** | 36.27 *** | 67.73 *** | 38.01 *** | 53.92 *** |

Note: Model 1 includes family structures (FSs), Model 2 includes FSs and selectivity factors, Model 3 includes FSs, selectivity, and stress, Model 4 includes all FSs, selectivity, and resources, and Model 5 includes all variables. Hispanics were not measured in the UK's data. Standardized coefficients reported in parentheses. * $p < 0.05$. ** $p < 0.01$. *** $p < 0.001$.

**Table 4.** Ordinary least squares regression of maternal depressive symptoms regressed on family structure, stress, resources, and controls in Australia (N = 3972).

|  | **Model 1** | **Model 2** | **Model 3** | **Model 4** | **Model 5** |
|---|---|---|---|---|---|
| Family Structure |  |  |  |  |  |
| Biological Cohabiting Stable | 0.141 * | 0.101 * | 0.043 | 0.043 | −0.007 |
|  | (0.048) | (−0.040) | (0.025) | (0.028) | (0.014) |
| Biological Single Stable | 0.621 *** | 0.455 ** | 0.322 * | 0.216 | 0.130 |
|  | (0.114) | (−0.088) | (0.067) | (0.046) | (0.034) |
| Post-Birth Biological Married | −0.001 | −0.069 | −0.121 | −0.099 | −0.136 |
|  | (0.008) | (−0.005) | (−0.018) | (−0.012) | (−0.021) |
| Post-Birth Stepfamily | 0.406 | 0.301 | 0.157 | 0.277 | 0.186 |
|  | (0.034) | (0.025) | (0.009) | (0.021) | (0.010) |
| Post-Birth Biological Cohabiting | 0.178 | 0.022 | −0.125 | −0.034 | −0.163 |
|  | (0.024) | (0.009) | (−0.007) | (0.003) | (−0.010) |
| Post-Birth Social Family | 0.275 * | 0.148 | −0.044 | 0.074 | −0.064 |
|  | (0.038) | (0.021) | (−0.014) | (0.009) | (−0.017) |
| Post-Birth Transition to Single | 0.481 *** | 0.425 *** | 0.297 *** | 0.213 * | 0.145 |
|  | (0.117) | (0.105) | (0.072) | (0.052) | (0.036) |
| Mother's Education |  |  |  |  |  |
| Secondary school |  | −0.012 | −0.037 | 0.039 | 0.006 |
|  |  | (0.003) | (−0.004) | (0.019) | (0.010) |
| Some college |  | −0.096 | −0.114 | −0.025 | −0.05 |
|  |  | (−0.026) | (−0.031) | (−0.008) | (−0.015) |
| Post-secondary degree |  | −0.116 | −0.133 | 0.002 | −0.027 |
|  |  | (−0.035) | (−0.039) | (0.004) | (−0.005) |
| Higher degree |  | −0.101 | −0.129 | 0.032 | −0.008 |
|  |  | (−0.031) | (−0.040) | (0.010) | (−0.004) |
| Parent's Immigration Status |  |  |  |  |  |
| One or more parent is an immigrant |  | 0.123 ** | 0.151 *** | 0.096 * | 0.127 * |
|  |  | (0.055) | (0.062) | (0.047) | (0.055) |
| Child Gender (1 = Male) |  | 0.016 | 0.020 | 0.012 | 0.013 |
|  |  | (0.000) | (0.000) | (−0.001) | (−0.002) |
| Child Age |  | 0.000 | −0.001 | 0.001 | 0.001 |
|  |  | (−0.005) | (−0.006) | (−0.002) | (−0.002) |
| Child Race |  | NA | NA | NA | NA |
| Mother's Age at Birth |  | −0.013 ** | −0.009 * | −0.009 | −0.007 |
|  |  | (−0.061) | (−0.044) | (−0.044) | (−0.035) |
| Family Stressor Scale |  |  | 0.235 |  | 0.237 *** |
| Income |  |  |  |  |  |
| Second |  |  |  | −0.208 ** | −0.161 * |
|  |  |  |  | (−0.060) | (−0.044) |
| Third |  |  |  | −0.208 ** | −0.156 * |
|  |  |  |  | (0.070) | (−0.051) |
| Fourth |  |  |  | −0.227 ** | −0.193 * |
|  |  |  |  | (−0.071) | (−0.056) |
| Top |  |  |  | −0.187 * | −0.153 |
|  |  |  |  | (0.062) | (−0.046) |

**Table 4.** *Cont.*

|  | **Model 1** | **Model 2** | **Model 3** | **Model 4** | **Model 5** |
|---|---|---|---|---|---|
| Moved |  |  |  | −0.007 | −0.111 * |
|  |  |  |  | (0.012) | (0.034) |
| State Support |  |  |  | 0.162 ** | 0.140 ** |
|  |  |  |  | (0.066) | (0.054) |
| Mother's Employment |  |  |  |  |  |
|   Part-time |  |  |  | −0.04 | −0.027 |
|  |  |  |  | (−0.032) | (−0.024) |
|   Not in paid labor force |  |  |  | 0.071 | 0.101 |
|  |  |  |  | (0.014) | (0.029) |
| Home Ownership |  |  |  | −0.079 | −0.055 |
|  |  |  |  | (−0.035) | (−0.026) |
| Household Size |  |  |  | −0.016 | −0.019 |
| Adjusted R-squared | 0.025 | 0.031 | 0.084 | 0.042 | 0.097 |
| F-statistic | 15.52 *** | 8.51 *** | 21.45 *** | 7.23 *** | 15.25 *** |

Model 1 includes Family Structures (FS), Model 2 includes FS and Stress, Model 3 includes FS, Stress, and Resources, and Model 4 includes all variables. Race is not measured in Australia data. * $p < 0.05$. ** $p < 0.01$. *** $p < 0.001$.

**Table 5.** Ordinary least squares regression of maternal depressive symptoms regressed on family structure, stress, resources, and controls in the United States (N = 8624).

|  | **Model 1** | **Model 2** | **Model 3** | **Model 4** | **Model 5** |
|---|---|---|---|---|---|
| Family Structure |  |  |  |  |  |
|   Biological Cohabiting Stable | 0.215 | 0.062 | 0.062 | 0.016 | 0.016 |
|  | (0.033) | (0.20) | (0.20) | (0.015) | (0.015) |
|   Biological Single Stable | 0.223 ** | 0.130 | 0.129 | 0.036 | 0.036 |
|  | (0.061) | (0.035) | (0.035) | (0.018) | (0.018) |
|   Post-Birth Biological Married | 0.087 | −0.021 | −0.021 | −0.056 | −0.056 |
|  | (0.021) | (−0.006) | (−0.006) | (−0.013) | (−0.013) |
|   Post-Birth Stepfamily | 0.177 | 0.111 | 0.110 | 0.069 | 0.069 |
|  | (0.024) | (0.014) | (0.014) | (0.010) | (0.010) |
|   Post-Birth Biological Cohabiting | 0.319 * | 0.171 | 0.170 | 0.103 | 0.104 |
|  | (0.038) | (0.019) | (0.019) | (0.012) | (0.012) |
|   Post-Birth Social Family | 0.251 | 0.185 | 0.184 | 0.124 | 0.125 |
|  | (0.024) | (0.014) | (0.014) | (0.006) | (0.006) |
|   Post-Birth Transition to Single | 0.100 * | 0.047 | 0.045 | −0.038 | −0.036 |
|  | (0.049) | (0.032) | (0.032) | (0.012) | (0.013) |
| Mother's Education |  |  |  |  |  |
|   Secondary school |  | −0.134 | −0.134 | −0.091 | −0.091 |
|  |  | (−0.039) | (−0.039) | (−0.012) | (−0.012) |
|   Some college |  | −0.218 * | −0.218 * | −0.142 | −0.142 |
|  |  | (−0.083) | (−0.083) | (−0.047) | (−0.047) |
|   Post-secondary degree |  | −0.336 *** | −0.336 *** | −0.215 * | −0.215 * |
|  |  | (−0.094) | (−0.094) | (−0.050) | (−0.050) |
|   Higher degree |  | −0.257 ** | −0.257 ** | −0.110 | −0.110 |
|  |  | (−0.065) | (−0.065) | (−0.025) | (−0.025) |
| Parent's Immigration Status |  |  |  |  |  |
|   One or more parent is an immigrant |  | 0.085 | 0.085 | 0.042 | 0.041 |
|  |  | (−0.003) | (−0.003) | (−0.019) | (−0.019) |
| Child Gender (1 = Male) |  | 0.064 * | 0.064 * | 0.063 * | 0.063 * |
|  |  | (0.011) | (0.011) | (0.013) | (0.013) |
| Child Age |  | −0.001 | −0.001 | −0.002 | −0.002 |
|  |  | (−0.013) | (−0.013) | (−0.015) | (−0.015) |
| Child Race |  |  |  |  |  |
|   Black |  | 0.033 | 0.034 | 0.006 | 0.006 |
|  |  | (0.021) | (0.021) | (0.013) | (0.013) |

**Table 5.** *Cont.*

| | Model 1 | Model 2 | Model 3 | Model 4 | Model 5 |
|---|---|---|---|---|---|
| Hispanic | | 0.151 ** | 0.151 ** | 0.113 * | 0.113 * |
| | | (0.080) | (0.080) | (0.069) | (0.069) |
| Asian | | −0.128 | −0.128 | −0.112 | −0.112 |
| | | (−0.020) | (−0.020) | (−0.020) | (−0.020) |
| Other | | 0.021 | 0.021 | −0.010 | −0.010 |
| | | (0.015) | (0.015) | (0.010) | (0.010) |
| Mother's Age at Birth | | −0.002 | −0.002 | 0.003 | 0.003 |
| | | (−0.002) | (−0.002) | (0.018) | (0.018) |
| Family Stressor Scale | | | 0.002 | | 0.002 |
| | | | (−0.002) | | (−0.002) |
| Income | | | | | |
| Second | | | | −0.099 | −0.099 |
| | | | | (−0.030) | (−0.030) |
| Third | | | | −0.138 * | −0.138 * |
| | | | | (−0.075) | (−0.075) |
| Fourth | | | | −0.226 *** | −0.226 *** |
| | | | | (−0.068) | (−0.068) |
| Top | | | | −0.283 *** | −0.283 *** |
| | | | | (−0.082) | (−0.082) |
| Moved | | | | 0.021 | 0.022 |
| | | | | (−0.010) | (−0.010) |
| State Support | | | | −0.056 | −0.056 |
| | | | | (−0.001) | (−0.001) |
| Mother's Employment | | | | | |
| Part-time | | | | −0.014 | −0.014 |
| | | | | (−0.009) | (−0.009) |
| Not in paid labor force | | | | 0.080 | 0.080 |
| | | | | (0.038) | (0.038) |
| Home Ownership | | | | −0.085 | −0.085 |
| | | | | (−0.016) | (−0.016) |
| Household Size | | | | 0.003 | 0.003 |
| Adjusted R-squared | 0.007 | 0.020 | 0.020 | 0.029 | 0.026 |
| F-statistic | 9.38 *** | 10.14 *** | 9.64 *** | 8.40 *** | 8.13 *** |

Note: Model 1 includes family structures (FSs), Model 2 includes FSs and selectivity factors, Model 3 includes FSs, selectivity, and stress, Model 4 includes all FSs, selectivity, and resources, and Model 5 includes all variables. Standardized coefficients reported in parentheses. * $p < 0.05$. ** $p < 0.01$. *** $p < 0.001$.

### 4.2. Model 2: Considering Selectivity Controls

In our first block model (Model 2 in each table), the following selectivity factors were included: mother's education level, parents' immigration status, child gender, child age, child race, and mother's age at birth. When controlling for these variables in Australia (Table 4) and the US (Table 5), the mothers in most family structures were predicted to have MDS levels that did not significantly differ from those of married stable mothers, with a few exceptions. In Australia (Table 4), the mothers in biological cohabiting stable ($b = 0.101$, $p < 0.05$), biological single stable ($b = 0.425$, $p < 0.001$), and post-birth transition to single families ($b = 0.455$, $p < 0.01$) were all expected to have significantly higher levels of MDS than mothers in married stable families. In the UK (Table 3), however, mothers in all family structures—except post-birth social families—continued to have expected MDS levels that were significantly higher than their married stable counterparts.

A few of the selectivity factors also presented interesting findings. For example, in Australia (Table 4), mothers who resided in families where at least one parent was an immigrant were predicted to have MDS levels that were significantly higher than those in families in which neither parent was an immigrant ($b = 0.123$, $p < 0.01$). Additionally, maternal education levels appeared to matter in the UK (Table 3) and the US (Table 5). In the UK (Table 3), mothers who completed secondary school ($b = −0.201$, $p < 0.001$), some college ($b = −0.289$, $p < 0.001$), a post-secondary degree ($b = −0.329$, $p < 0.001$), or a

higher degree ($b = -0.240$, $p < 0.001$) were expected to have MDS levels that were lower than those for mothers who completed less than secondary school. Similarly, in the US (Table 5), mothers who completed some college ($b = -0.218$, $p < 0.05$), a post-secondary degree ($b = -0.336$, $p < 0.001$), or a higher degree ($b = -0.257$, $p < 0.01$) were expected to have lower MDS levels than mothers who completed less than secondary school.

*4.3. Model 3: Considering Stress*

In Model 3, the family stressor scale proved to be an important predictor of MDS levels in the UK only (see Table 3). Mothers in the UK who experienced greater stress as captured by this scale were predicted to have higher levels of MDS ($b = 0.193$; $p < 0.001$), while no significant relationship between MDS and stress was found in Australia (Table 4) or the US (Table 5). Additionally, the selectivity factors previously discussed in Model 2 continued to follow similar patterns in the presence of the stressor scale. In each country, however, the presence of the stressor scale had differing effects on whether mothers in alternative family structures continued to report MDS levels that were significantly higher than those for married stable mothers.

In Australia, only two family structures continued to have MDS levels that were significantly higher than those for married stable mothers (Table 4, Model 3). When controlling for the family stressor scale, Australian mothers who resided in a biological single stable family ($b = 0.322$, $p < 0.05$) or a post-birth transition to a single family ($b = 0.297$, $p < 0.001$) were both predicted to have MDS levels that were significantly different from the married stable comparison group. In the UK (Table 3, Model 3), five of the family structures continued to be significantly different from the married stable comparison group, with mothers in biological cohabiting stable ($b = 0.095$, $p < 0.05$), post-birth biological married ($b = 0.289$, $p < 0.01$), post-birth stepfamily ($b = 0.229$, $p < 0.01$), post-birth biological cohabiting ($b = 0.192$, $p < 0.01$), and post-birth transition to single ($b = 0.322$, $p < 0.001$) families continuing to have MDS levels that were significantly higher than their married stable counterparts. These findings in Australia and the UK indicate that the differences in the MDS levels between family structures generally could not be accounted for by the stressor scale. All family structures in the US continued to have MDS levels that were not significantly different from the married stable group (Table 5, Model 3).

*4.4. Model 4: Considering Resources*

In our fourth model, we removed the stressor scale and added our variables capturing a family's access to resources. These variables included income, residential mobility, state support, mother's employment, home ownership, and household size. With the introduction of these variables in each country, several family structures were no longer predicted to have significantly different MDS levels. In Australia (Table 4), only mothers in post-birth transition to single families continued to have MDS levels that were expected to be significantly higher than married stable mothers ($b = 0.213$, $p < 0.05$). In the US (Table 5), all family structures continued to have MDS levels that did not significantly differ from mothers in married stable families. In the UK (Table 3), however, half of all family structures continued to have significantly higher MDS levels. Specifically, mothers in the UK in post-birth biological married ($b = 0.106$, $p < 0.01$), post-birth stepfamily ($b = 0.196$, $p < 0.05$), post-birth biological cohabiting ($b = 0.134$, $p < 0.05$), and post-birth transition to single ($b = 0.176$, $p < 0.001$) families were expected to have MDS levels that were significantly higher than those of mothers in married stable families.

A few of these resource variables also presented interesting findings. Across all three countries, income had a negative association with MDS, in that mothers with higher incomes were predicted to have lower MDS. This was true of all quintiles in Australia, the three top quintiles in the US, and the two top quintiles in the UK. Additionally, families in Australia and the UK who received income support from the government reported higher levels of MDS than mothers who did not (AUS: $b = 0.162$, $p < 0.01$; UK: $b = 0.296$, $p < 0.001$).

*4.5. Model 5: Considering How Resources and Demographics Modify Stress*

Our final model (Model 5) included selectivity factors, the stressor scale, and the resource variables to assess how the stressors affected mothers in different family structures when taking into account resources and demographics. When controlling for these variables in Australia (Table 4) and the US (Table 5), the mothers in all family structures were predicted to have MDS levels that did not significantly differ from those of married stable mothers. This essentially replicated the findings from previous models for Australia, where no family structures exerted significant effects in the presence of key controls. When controlling for these variables in the UK (Table 3), however, mothers in post-birth biological married families ($b = 0.097$, $p < 0.01$), post-birth stepfamilies ($b = 0.154$, $p < 0.05$), and post-birth transition to single families ($b = 0.176$, $p < 0.001$) were all expected to have significantly higher levels of MDS than mothers in married stable families. The resource variables identified in Model 4 continued in a similar pattern, wherein a lower income and heightened stress were associated with increased MDS levels. One possible interpretation of these results is that there is something about family formation after marriage in the UK that is distinct. Another possible interpretation is that there is something distinct in the US context about how resources ameliorate the effects of family structure, an interpretation perhaps not surprising given the dearth of social safety nets in the US.

## 5. Sensitivity Tests

To ensure the robustness of our findings, we also considered comparisons between each possible pair of family structures in each country (for example, between post-birth stepfamilies and post-birth social families). Across these 168 comparisons, we found only two additional significant relationships (AUS: post-birth bio married families had higher MDS than post-birth transition to single families; UK: stable cohabitors had higher MDS than post-birth transition to single families). These were not consistent across countries nor compared to the findings discussed above, so we did not pursue them further. Additionally, the life course perspective indicates that the number of transitions may have been important for understanding mental health trajectories, so we performed further analysis exploring the number of transitions as an indicator for family instability. We found similar patterns to those discussed above; however, these results were not included in our models, as they introduced multicollinearity and suggested similar trends concerning instability to the models we reported above.

We urge the reader to examine the standardized regression coefficients, presented in parentheses below the unstandardized regression coefficients, in tandem with the hypothesis tests (Lash 2017). Using just null hypothesis significance testing, we found that no family structure in the US or AUS was related to MDS when other explanatory variables were included. However, the sample sizes in the US and AUS were smaller than that in the UK. If we examine the predicted values of MDS (Figure 1), we see smaller differences in terms of real-world effects than we might assume from the statistical significance tests reported above. In addition, perhaps because of a smaller sample size and commensurately smaller family structure groups in the Australian data, the confidence intervals for Australia were much larger than those for the UK and US data. While our sample sizes were still sufficiently large to consider statistical significance, we encourage readers to consider the significance in conjunction with effect sizes and note that we are most confident in the differences between post-birth single-parent families in the UK and the US. Data collected by the same organization that included all three countries might allow for multilevel models that can help adjudicate these questions in the future (see Limitations section below).

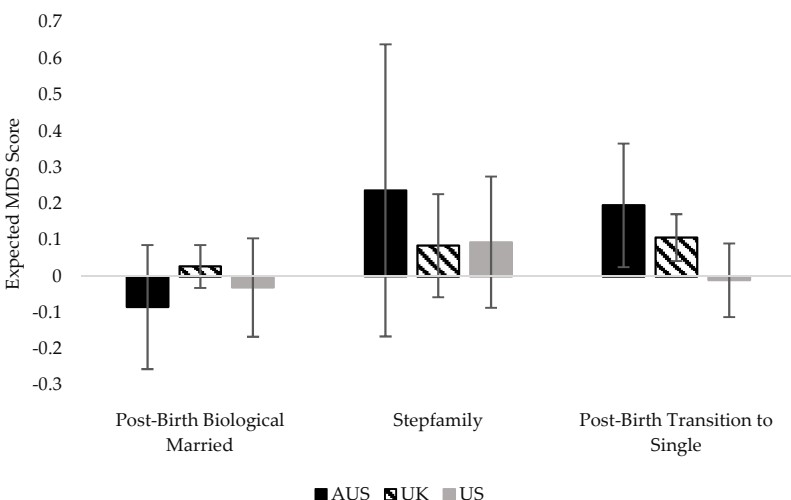

**Figure 1.** Expected MDS score by family structure in Australia, the US, and the UK.

## 6. Discussion

We set out to examine how the stress process model might help to explain differences in maternal depressive symptoms across two contexts: eight different family structures and three different countries. While selectivity factors, resources, and stress were associated with MDS, we find it noteworthy that the family structure may influence MDS differently in the UK than it does in Australia or, especially, in the US. Whereas selectivity factors, resources, and stress largely accounted for the link between family structure and MDS in Australia and the US, three family structures in the UK remained directly associated with MDS, though we note that comparing the effect sizes showed more notable differences for those family structures for mothers in the UK compared with the US and perhaps minimal differences for mothers in the UK compared with Australia. Mothers in the UK context who resided in post-birth biological married families, post-birth stepfamilies, and post-birth transition to single families had higher average MDS levels than mothers in stable married families, even when we considered their selectivity factors, exposure to stress, and their access to resources.

These three family structures are all the result of a mother either entering into or dissolving a legal contract after the birth of their child. This demonstrates that there could be cultural differences surrounding marriage patterns and associated parental pathways in the UK that do not exist in the US and Australia, possibly linking the legal processes and cultural norms of marriage in the UK to rates of MDS. While some of the other family structures in the UK may be protected against the heightened strain of parenting through social policies that support families and children, it appears there may be a unique strain associated with entering or leaving a marriage in the UK that we are not capturing and that social policies do not address. Additionally, the access to family-supportive policies may provide the impetus and efficacy to leave or enter marriages in the UK, regardless of the related maternal depressive symptoms that may result. We suggest further research into the context of the UK's culture and legal processes, as a more in-depth exploration of this pattern could provide insight into how the institution of marriage influences the rates of MDS in the UK. At the same time, it seems clear that maternal mental health in the US was more sensitive to resources—or the lack thereof—than was true in the Australian or UK samples. The unique characteristics of the US political landscape and the corresponding lack of social safety nets may be to blame for this pattern.

We also expected selectivity factors, resources, and stress to be related to this link between the family structure and maternal depressive symptoms. Our models provided evidence that selectivity factors such as immigration and level of education, as well as limited access to resources and heightened stress, were associated with higher levels of MDS in all three of the countries we examined here. The stress process model provides an

explanation for this through its inclusion of resources as an important aspect of the stress process. Certain stressors, such as losing a job, can influence how many resources a family has access to, which in turn is associated with a mother's likelihood of exhibiting MDS. Additionally, the resources a mother has can either mitigate or exacerbate the effect that exposure to stress can have on their mental health (McLeod 2012; Pearlin et al. 1981). For example, although we may expect a mother who transitioned into being a single parent to have significantly more depressive symptoms than a stably married mother, our results indicate that this likely has less to do with the family structure itself and more to do with increased role strain, changes in trajectory, and a financial inability to cope with being a single parent (Nomaguchi and Milkie 2020). We also speculated on a potential positive effect for mothers whose pathway changes through entering into a new union and gaining access to new resources. However, we found no evidence that a mother who enters into a union experiences lower MDS levels. While there are questions about how resources in blended families might have to be shared with family members who reside elsewhere, the fact that entering a union that presumably brings with it more resources is not associated with lower MDS levels tells us that the links among family instability, stress, resources, and maternal mental health are complex. While we found some notable differences across contexts—particularly between the UK and the US—the consistent findings that stress and resources were linked to MDS indicate that though some research might presume family structure to be a driving force in understanding family processes and outcomes, factors related but external to the family structure itself may play a much larger role.

## 7. Limitations

In this cross-national comparison, one of the main limitations we faced was making direct comparisons across data sets. Although we made it a priority in our data preparation to harmonize variables and concepts to allow for true comparisons, this was not always feasible. For example, each stressor scale ultimately captured different facets of stress in each country, making direct comparisons difficult. Likewise, two different depression indices were used across three countries. To accommodate differences in the measures, we standardized the scales so that direct comparisons could be made. In addition, we could not assess race as a confounding factor in Australia because race and ethnicity were not asked about in the data. In addition, we could not conduct multilevel models with the country as the second level as a direct test of whether slopes on key variables differed across countries because of how the data were gathered.

Because of the focus on maternal outcomes in these datasets, one major limitation is the failure to note previous transitions in mothers' lives prior to the birth of their child. Another potential limitation is the relatively small proportion of mothers who reside in some of the alternative family structures, especially in the smaller Australian sample. One concern with measuring family structure so finely is whether there are enough respondents in each of these categories. While additional statistical tests here indicated a sufficient cell size, data with larger samples of alternative families could provide an even more nuanced test of the questions we explored here.

We also note that the data we used were unable to distinguish between child care support among a possible set of social safety net supports from governments in these countries. Additional tests on whether different kinds of support in the one country broke out government support into additional categories (the UK) did not show any substantive differences in the availability or use of different kinds of support, but it is possible that more detailed variables tapping support that specifically eases parenting burdens might be associated with both better maternal mental health and with different slopes for the effects of support in different family structures. The data were further unable to determine support, either monetary or in terms of shared parenting, from non-residential parents. Again, we might assume mothers in alternative family structures with more access to such non-residential parent resources might report better mental health. Still, the data provided a number of strong measures of family mechanisms and resources that allowed

confidence in the findings we reported here. Finally, because of the nature of our data, we were unable to examine these questions for mothers parenting with another woman, as there were too few mothers in such relationships in these data to make useful estimates, but we acknowledge there may be important similarities and differences in the associations between the family structure and maternal mental health for same-sex couples.

## 8. Implications

Our findings indicate that selectivity factors, resources, and exposure to stress are important predictors in understanding maternal mental health across multiple contexts, and policymakers should focus on these as avenues to addressing MDS. Considering the negative ramifications that MDS can have on the larger family unit, our research provides evidence that limiting the mothers' exposure to stress and increasing their access to resources can help benefit not only maternal mental health but the family system as a whole (Burke 2003; Goodman et al. 2011). We therefore recommend that policymakers closely examine their economic safety nets and seriously consider the mental health of mothers in low-income families when passing policies to provide economic relief. Resources devoted to encouraging marriage as a solution to family and child problems could be diverted to providing economic and mental health support across a variety of family structures. Additionally, we suggest that future research focus on specific cultural norms and policy differences across macro-level contexts and how those differences might impact the association between the family structure and MDS. We suggest that a deeper exploration of contextual differences in regard to union formation and dissolution may be able to further explain the differences in the association between the family structure and MDS that we found between the UK and the US. As a result, policies need to be tailored to culturally specific audiences. Ultimately, country context matters, and resources are an important attenuating factor in the association between family structure, stress, and maternal depression.

**Author Contributions:** Conceptualization, J.A.J., M.J.D. and S.P.; formal analysis, K.R., E.K.S. and S.A.S.; methodology, J.A.J., M.J.D. and S.P.; supervision, M.J.D.; visualization, K.R.; writing—original draft, K.R., E.K.S. and S.A.S.; writing—review and editing, K.R., J.A.J., M.J.D. and S.P. All authors have read and agreed to the published version of the manuscript.

**Funding:** This research received no external funding.

**Institutional Review Board Statement:** Not applicable.

**Informed Consent Statement:** Not applicable.

**Data Availability Statement:** Restrictions apply to the availability of these data. Australian data was obtained from the Australian Bureau of Statistics and is available from https://growingupinaustralia.gov.au/data-and-documentation/accessing-lsac-data, accessed on 7 May 2020. UK data was obtained from the Centre for Longitudinal Studies and is available from https://beta.ukdataservice.ac.uk/datacatalogue/series/series?id=2000031, accessed on 7 May 2020. US data was obtained from the National Center for Education Statistics and is available from https://nces.ed.gov/ecls/kinderdatainformation.asp, accessed on 7 May 2020.

**Acknowledgments:** We thank Kristie Rowley, who was instrumental in starting the larger International Family Project at BYU. We also thank Yuanyuan Yue and Alex Wambach for their preliminary work on this paper.

**Conflicts of Interest:** The authors declare no conflict of interest.

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
