# Peer review of "Family Structure and Maternal Depressive Symptoms: A Cross-National Comparison of Australia, the United Kingdom, and the United States"

_socsci, doi:10.3390/socsci11020078_

Round 1

Reviewer 1 Report

Thank you for the opportunity to review “Family Structure and Maternal Depressive Symptoms: A Cross-national Comparison of Australia, the United Kingdom, and the United States” [Manuscript ID: socsci-1468429]. This manuscript presents a study consolidating data across three countries to assess the association between family structure and maternal depressive symptoms. This study contributes to the literature regarding maternal depressive symptoms outside of the postpartum period.

Introduction

Lines 24 and 40 need a citation.

Background

Throughout the introduction, the authors reference “families” in general but are not specific about whether they refer to heterosexual married couples, blended families and whether the literature differs concerning these types of unions. Lines 103-119

I suggest the authors describe similarities/ differences across these country contexts for average age at first union, divorce/ separation rates and any other important demographic differences that should be considered in advance of the analysis. For example, different countries have different norms with regard to childbirth in non-marital unions. The authors do a nice job of pointing out differences in structural supports/ resources that vary across country contexts.

Line 86 the authors introduce two applications of the stress process model: 1) family structure and 2) country context and then go on to discuss country context first which is confusing to the reader. Please reorder for clarity.

Lines 132-149

Measures, family structure: please be clear that same-sex partners/ families are excluded from the analysis or consider grouping (or running a sensitivity analysis) on non-biological partners/ parents who may be part of a stable domestic partnership or union.

Measures, stress – large differences in the way stress was measured across country contexts are problematic for analysis. For this reason, and others mention below, it seems advisable to have a more limited role for these variables (e.g., running sociodemographic-adjusted models without stress measures before adding stress measures).

Methods, Data lies 231-234: “For all data sets, we limit our sample to mothers for whom we could identify reliable measures of family structure across waves; this results in a total sample size of 3,972 mothers from Australia, 12,821 mothers from the UK, and 8,624 mothers from the US.” – how many excluded due to “unreliable” measures? What were unreliable measures?

What is the rationale for including stress measures? As confounders or mediators? If the latter (which I think is the stronger argument), then they should not be included in the first set of adjustment variables, but rather be added after the influence of potential confounders such as sociodemographics can be assessed.

Likewise for other financial measures – are these conceptualized as confounders or mediators? They could reasonably be conceptualized as either or both.

Table 1 - How was income measured? Was quintile based on entire sample distribution?

Analysis

Analysis - Were all country samples included in a single regression model? How did authors handle missing race data for Australia when controlling for demographics?

Is there any longitudinal data on MDS available? If not, I think it is important for the authors to be clear that this study cross-sectional, since there is no information on whether MDS preceded or followed changes in family structure.

Ordering of nested models should be first crude then add demographics, then add potential “explanatory” variables such as stress and resources. Otherwise it is impossible to know impact of confounders on effect estimates. (In particular, it is not possible to know how important race/ethnicity confounding is for UK/US, which may help to understand if lack of race/ethnicity data in Australia may be an important source of differences in parameter estimates across countries.)

Many effect sizes for family structure are as large in US and Australia as they are in UK, but are not statistically significant likely due to smaller sample. The authors interpret non-significant results as null, which is problematic because it conflates evidence for the null (parameter estimate near 0 and tight confidence intervals) with lack of evidence against the null (large parameter estimate with wide confidence intervals). Authors should interpret strength of parameter estimates rather than solely relying on null hypothesis significance testing, as relying solely on p-values may lead to incorrect interpretations of differences across countries. It would also help authors make clear when parameter estimates provide evidence for the null and when parameter estimates suggest a non-null effect but with imprecision resulting in inclusion of the null in the 95% CIs. Authors are encouraged to review the following paper on p-values and statistical testing. Timothy L. Lash, The Harm Done to Reproducibility by the Culture of Null Hypothesis Significance Testing, American Journal of Epidemiology, Volume 186, Issue 6, 15 September 2017, Pages 627–635, https://doi.org/10.1093/aje/kwx261

Limitations, inability to harmonize data stress indicators across datasets, and effectively measuring different types of stressors is problematic for the overall interpretation of results.

Discussion – possibility to discuss how access to resources and social safety-net services may influence self-efficacy to initiate a change in family structure (formation/ dissolution) and therefore contribute to measures of MDS.

The authors’ conclusion that “Whereas resources and stress largely explain the link between family structure and MDS in Australia and the US, three family structures in the UK remain directly associated with MDS” is not supported by the results, and relies on significance testing, which conflates strength of association with sample size. (The UK sample size is much larger than the others, and the US samples size is by far the smallest; these differences in sample size will mean that the same size effect estimate is statistically significant in the UK but not in the other countries.)  If one compares parameter estimates, rather than p-values, across countries, it is not at all clear that the UK is the outlier in terms of impact of adjustment for resources.  Rather, it looks more like the US is the outlier in that resources seem to “explain” far more of the effect of the association of family structure with MDS.  A different presentation of results is needed to more fairly present the nuances found here. The authors might consider presenting a figure of parameter estimates with 95% CIs for each family structure by country. This would make it easier to see how different or similar the effect estimates are across country contexts.

Reviewer 2 Report

Accept with minor changes. Do not use personal language in academic writing. Include confidence intervals in the results section.

Reviewer 3 Report

In my opinion, the manuscript is overall well written and deals with an important topic. The authors did a nice job summarizing previous studies available, as well as in the discussion of the results.

Below please consider the minor points to improve the text:

A significant limitation of the study is the question of measuring depression symptoms: if different measuring instruments were used in different countries, this could have a significant effect on the results- this needs to be mentioned in the limitations paragraph of the study. However, it is not clear how did authors proceed in the “construction of the scale of depressive symptoms using a series of mental health questions” (line 236)- how was this done? Did the authors select some of the questions from the scales, or did they combine questions from more measuring instruments (if so, which instruments?)- for instance in US data- did authors use only the questions from the Kessler Psychological Distress Scale, or also from CES-D? This needs to be clarified in the methods section of the manuscript.

Moreover, it would be useful to add more information on the validity and reliability of the measuring instruments used in different countries. (Kessler Psychological distress scale did not measure depression, but general, non-specific psychological distress…)

It would be useful to add the references to Table 1, Table 2, Table 3 in the Results section of the manuscript, and this section could be divided into subheadings to increase the readability and clarity of the text.

Round 2

Reviewer 1 Report

I appreciate the additional detail on modeling decisions and results. My concerns have been addressed.

Author Response

Thank you for your review and for your additional suggestions that have helped improve the readability and accuracy of the paper. We have made the requested line edits and removed the language in lines 127-132 about stepmothers. We agree with the issue editor on the issue of “personal language” and have therefore elected to not remove the personal language from our manuscript (as R1 suggested) in favor of using active voice to be more concise. All track changes and highlighting have been removed from the current draft.  

Thank you again for your careful attention to this paper and for the opportunity to publish in Social Sciences. Please let us know if there are any additional changes or if there is any additional information we can provide.